# Manufacturing via Plasma Metal Deposition of Hastelloy C-22 Specimens Made from Particles with Different Granulometries

Isabel Montealegre-Meléndez [1], Eva M. Pérez-Soriano [1,*], Enrique Ariza [1,2], Erich Neubauer [2], Michael Kitzmantel [2] and Cristina Arévalo [1]

1. Escuela Politécnica Superior de Sevilla, Universidad de Sevilla, Calle Virgen de África 7, 41011 Sevilla, Spain; imontealegre@us.es (I.M.-M.); e.ar@rhp.at (E.A.); carevalo@us.es (C.A.)
2. RHP-Technology GmbH, Austrian Research Center, 2444 Seibersdorf, Austria; e.ne@rhp.at (E.N.); m.ki@rhp.at (M.K.)
* Correspondence: evamps@us.es; Tel.: +34-954-55-43-57

**Abstract:** Additive manufacturing techniques offer significant advantages for creating complex components efficiently, saving both time and materials. This makes them particularly appealing for producing parts from intricate alloys, such as Hastelloy C-22. One such technique, plasma metal deposition, uses plasma on powdered material to build up layers. The novelty of this work is to analyze and determine whether there is a correlation between the particle size and the final behaviour of specimens produced via additive manufacturing. To achieve this, four powders with an identical chemical composition but different granulometries were employed. Additionally, some of the samples underwent thermal treatment (progressive heating at 10 °C/min until 1120 °C, maintained for 20 min, followed by rapid air cooling). Four walls were constructed, and after mechanical, tribological, and microstructural characterization, it was determined that the influence of the thermal treatment remained consistent, regardless of particle size. It was observed that the particle size slightly affected the final properties: the finer the powder, the lower the ultimate tensile strength values. Furthermore, it was evident that the thermal treatment substantially affected the microstructure and wear behavior of all the specimens, regardless of their initial particle size.

**Keywords:** Hastelloy C-22; plasma metal deposition; mechanical properties

## 1. Introduction

Over the past few decades, advancements in layer-by-layer manufacturing have proven pivotal in developing innovative fabrication methods for producing specimens from specialized metals and alloys with intricate geometries [1–6]. The advantages offered by additive manufacturing (AM) techniques make them highly attractive and in-demand in the industrial sector. Notably, substantial savings in both raw materials and fabrication time distinguish these techniques from traditional manufacturing processes [7–9].

There is a wide variety of AM techniques available, with some of them currently being implemented in the industry. Classifications of these techniques can be made in different ways according to the energy employed during the process or considering the deposition and feed of the raw materials [10]. Layer-by-layer manufacturing of metals requires a laser or plasma as an energy source. When the energy is supplied at the same time as the raw material, this process is called direct energy deposition (DED) [11–13]. Plasma metal deposition (PMD) is included in this group of techniques, in which the source of energy is generated by an arc. The peculiarity of this process lies in the feed of the raw materials, since they can be in a powder or wire form. It allows for the creation of complex components with a high output volume, promoting the sustainability of this technique [14,15].

In the framework of metals, Hastelloy C-22 is a nickel-based austenitic alloy that also contains chromium, molybdenum, iron, and cobalt [16]. It offers excellent corrosion resistance, in addition to its good mechanical properties. Among its properties, it can

include a high resistance to pitting and crevice corrosion, in addition to stress corrosion cracking [17]. On the one hand, its high chromium content ensures a good resistance in oxidizing media, while on the other hand, molybdenum and tungsten provide resistance in reducing media. Moreover, it has excellent resistance in oxidizing aqueous media, including liquid chlorine, as well as in solutions with nitric acid [18,19]. Due to this environmental resistance, this material is widely used in various sectors, such as chemical, petrochemical, aerospace, health, etc. [20–24].

This work focuses on investigating Hastelloy C-22 specimens produced via PMD with the aim of deepening the current knowledge on this material regarding its composition, microstructure, mechanical, and tribomechanical properties. It seeks to provide answers to what the optimal manufacturing parameters are, depending on the requirements. Furthermore, since the raw material employed is Hastelloy C-22 powder, the particle size of the starting material (granulometry) has been studied as an influencing factor on the final properties of the specimens. In order to achieve mechanical properties according to the standards, thermal treatments have been carried out. In this regard, the effects of these treatments on the microstructure, as well as on the mechanical and tribomechanical properties of the specimens, are explored.

## 2. Materials and Methods

As previously mentioned, Hastelloy C-22 is a versatile austenitic alloy. The starting material used in this research was supplied in powder by the company Atomizing Systems Limited (Sheffield, UK), which was manufactured using the plasma atomization process. This process involved melting and atomizing the material into fine droplets using an argon plasma torch as the heat source. The compositions of the standard Hastelloy C-22 [25] and the material used in this research are presented in Table 1.

**Table 1.** Weight compositions of the material studied and the standard alloy.

| | Weight by Weight Composition [%] | | | | | | | | | |
|---|---|---|---|---|---|---|---|---|---|---|
| | Ni | C | Co | Cr | Fe | Mn | Mo | Si | V | W |
| **Hastelloy C-22 studied** | Bal. | 0.007 | 0.40 | 22.11 | 2.00 | 0.65 | 12.37 | 0.91 | <0.02 | 3.63 |
| **Hastelloy C-22 standard** | Bal. | <0.015 | <2.50 | 20–22.5 | 2–6 | <0.50 | 12.5–14.5 | <0.08 | <0.35 | 2.5–4 |

The starting material was sieved, and three ranges of powder size were obtained for further study: $P_1$, fine powder; $P_2$, medium powder; and $P_3$, coarse powder. The original unsieved powder, $P_O$, was also included in the study. Table 2 presents the nomenclature used to identify the different powders and walls, along with the sizes $d_{10}$, $d_{50}$, and $d_{90}$ ($d_{10}$ denotes the particle diameter at which 10% by weight of the particles are smaller than this diameter, while $d_{50}$ and $d_{90}$ provide analogous measurements for 50% and 90%). These values were measured using a Mastersizer 2000 particle size analyzer (Malvern Instruments Ltd., Malvern, Worcestershire, UK). Figure 1 displays secondary electron–scanning electron microscopy (SE-SEM) images that reveal the spherical morphology of the powder particles from the starting material, Hastelloy C-22.

**Table 2.** Identification and information of the different granulometries obtained, including their $d_{10}$, $d_{50}$, and $d_{90}$ values.

| Wall ID | Granulometry [μm] | $d_{10}$ [μm] | $d_{50}$ [μm] | $d_{90}$ [μm] |
|---|---|---|---|---|
| $P_O$ | 50–175 | 57.39 | 82.74 | 125.14 |
| $P_1$ | 50–80 | 48.63 | 66.03 | 89.55 |
| $P_2$ | 80–125 | 57.75 | 80.71 | 112.11 |
| $P_3$ | >125 | 91.99 | 126.27 | 171.79 |

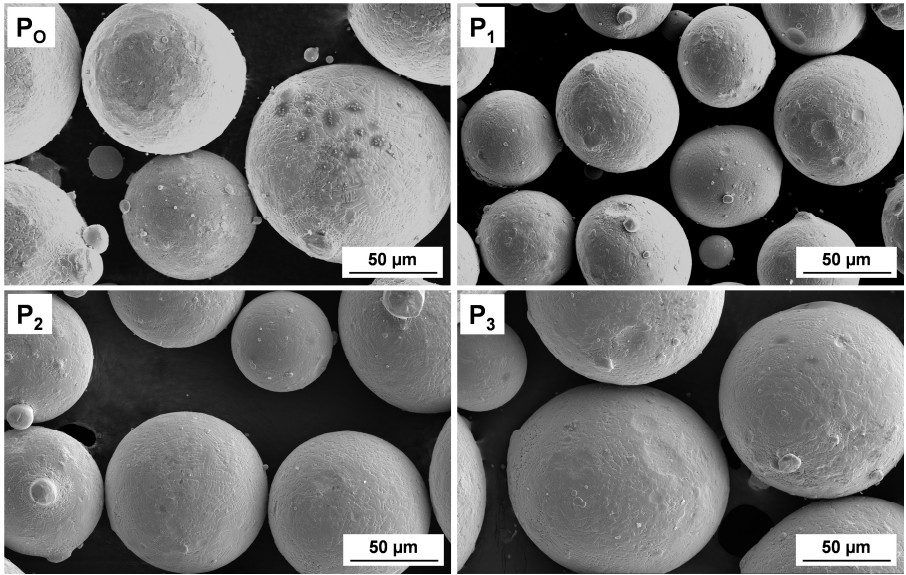

**Figure 1.** SE-SEM images of the starting raw material Hastelloy C-22 in its original batch ($P_O$), as well as in the form of a fine powder ($P_1$), medium powder ($P_2$), and coarse powder ($P_3$).

Using these powders, along with the unscreened powder obtained upon reception, four walls of Hastelloy C-22 were manufactured via PMD by the company RHP-Technology (Seibersdorf, Austria). The schematic illustration of the PMD process is depicted in Figure 2.

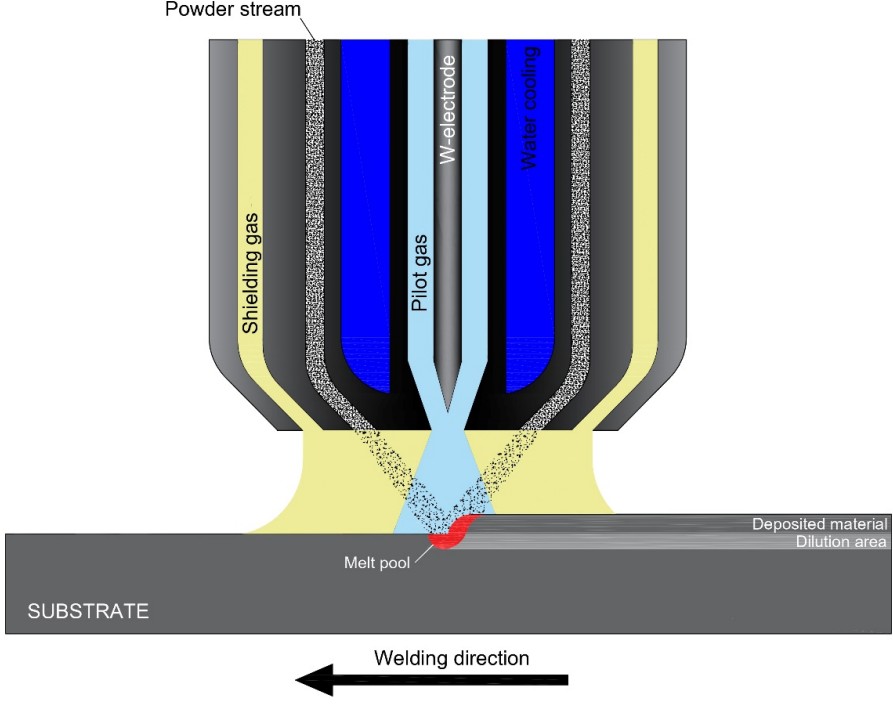

**Figure 2.** Diagram of the PMD equipment.

The manufacturing process involved several parameters that were selected from previous authors' research [26]. The energy source operated at a current intensity of 130 A, while the torch was programmed to move at a speed of 700 mm/min with oscillating movements. The deposition flow rate was 22.5 g/min. For improving the powder flow, argon was injected (powder gas) with a flow rate of 1.5 L/min. During the process, the distance between the torch and the built wall was 10 mm. Additionally, argon was used as a shielding gas, with a 99.99% purity and a flow rate of 15 L/min. This gas was introduced

around the plasma arc to locally protect the weld seam from oxidation or other external agents that could enter during the manufacturing process. These parameters remained identical for the four walls. The resulting samples can be observed in Figure 3.

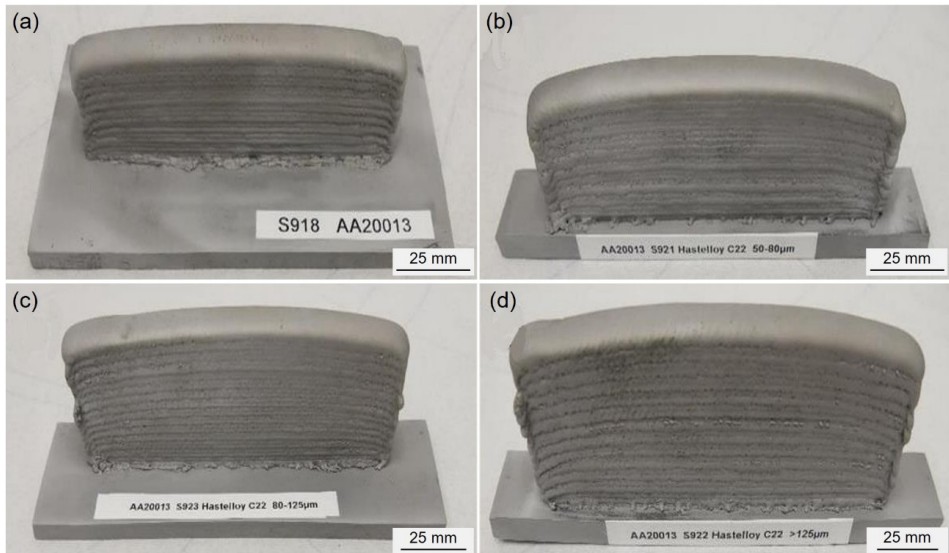

**Figure 3.** Images of the Hastelloy C-22 manufactured samples: (**a**) wall with original powder, $P_O$; (**b**) wall with fine powder, $P_1$; (**c**) wall with medium powder, $P_2$; (**d**) wall with coarse powder, $P_3$.

After the walls were manufactured, they were machined by wire cutting to obtain specimens with a normalized geometry for the tensile tests [27].

The extracted specimens were divided into two series. The first series remained as-built (*A*), while the second series received an additional thermal treatment (*T*). This thermal treatment involved progressive heating within a furnace at a rate of 10 °C/min until a temperature of 1120 °C was reached. This temperature was maintained for twenty minutes, followed by rapid air cooling (RAC). The *A-x* series refers to specimens that have not been subjected to thermal treatment, where *x* represents the specimen number in this series. On the other hand, the *T-x* series corresponds to specimens that have been treated. The thermal treatment and nomenclature of the samples are summarized in Table 3.

**Table 3.** Characteristics of the thermal treatment cycle and nomenclature of the samples studied.

| Condition | Thermal Treatment Cycle | Samples |
|---|---|---|
| As-built | - | A-1, A-2, A-3, A-4 * |
| Thermal treatment | 10 °C/min, 1120 °C, 20 min, RAC | T-1, T-2, T-3, T-4 * |

\* Specimen 4 is not present in every wall.

The number of specimens extracted from each wall was variable. The position and location of each sample on the walls, indicating which received thermal treatment they received, is presented in Figure 4. For the analysis of the influence of thermal treatment, specimens extracted from similar positions with respect to the substrates were compared.

The microstructural study was carried out after a metallographic preparation, using a Nikon optical microscope Eclipse MA100N (Nikon, Tokyo, Japan) and a scanning electron microscope (SEM, FEI Teneo, Hillsboro, OR, USA), equipped with an EDS (X-ray energy dispersive spectrometer) for composition analysis and elemental mapping.

The X-ray powder diffraction (XRD) analysis was conducted using a Bruker D8 Advance A25 (Billerica, MA, USA). Cu-K$_\alpha$ radiation was employed for phase characterization. A reference intensity ratio (RIR) analysis was employed to semi-quantitatively determine the present phases.

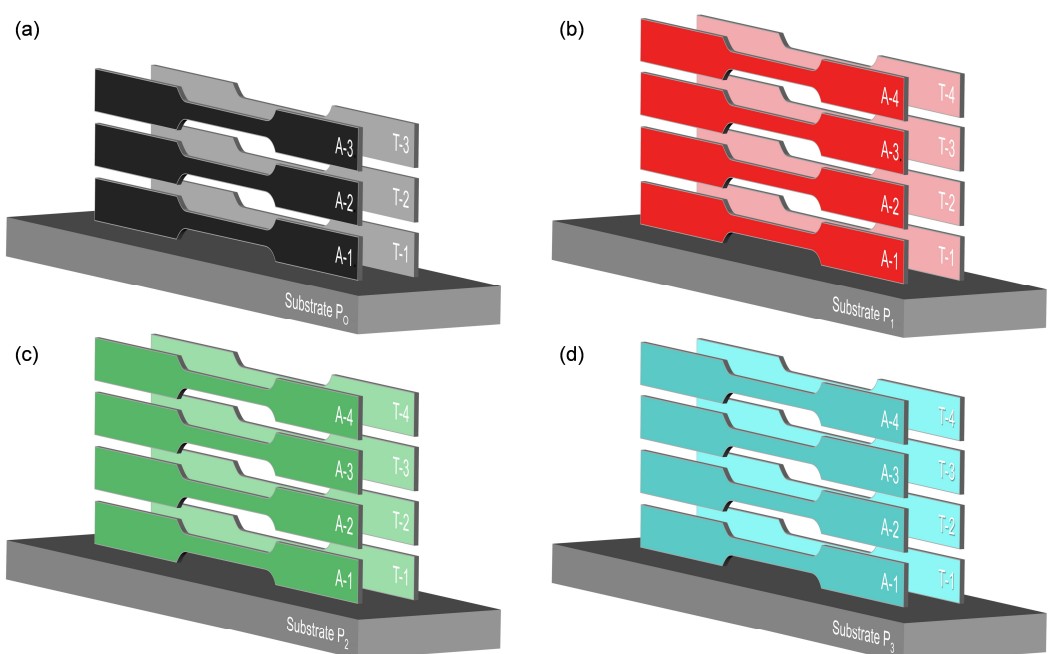

**Figure 4.** Representation of specimens extracted from each wall: (**a**) original powder, $P_O$; (**b**) fine powder, $P_1$; (**c**) medium powder, $P_2$; (**d**) coarse powder, $P_3$.

A study of the mechanical properties was performed using tensile tests [27], applying the load longitudinally to the specimen using an Instron 5505 equipment (Norwood, MA, USA) with a deformation speed of 0.5 mm/min at room temperature. Based on these tests, the ultimate tensile strength (*UTS*), the yield strength (*YS*), and the Young's modulus (*E*) values were obtained.

To determine the microindentation values, a Shimadzu HMV-G MicroVickers Hardness Tester (Shimadzu, Kyoto, Japan) was used. Three tests were carried out, applying a progressive increase in load until the final value of the maximum load was reached up to 1 N, 5 N, and 10 N for one minute, respectively. These values were maintained for 40 s, then the load was released to study the elastic recovery produced.

The tribological test was performed at room temperature (relative humidity in the range of 30–35%) using a ball-on-disc Microtest MT/30/NI tribometer (Microtest, Madrid, Spain). The ball of the tribometer had a 6 mm diameter, was alumina, and was applied with a force of 5 N for 15 min. The radius of the resulting groove was 2 mm, and the rotational speed was set to 200 rpm.

## 3. Results

### 3.1. Microstructural Analysis and DRX

The microstructural analysis unveiled a homogeneous matrix containing a minor degree of microporosity. It exhibited a spherical morphology that was uniformly distributed throughout the matrix. This observation is evidenced in the microstructure images depicted in Figure 5, captured via optical microscopy (OM). These images are categorized according to the starting powder utilized ($P_O$, $P_1$, $P_2$, and $P_3$) and the specimen's condition, either as-built (A) or subjected to thermal treatment (T).

The matrix is identified as a light grey color, comprising columnar grains attributed to the rapid cooling rate experienced during manufacturing. Additionally, segregation of the precipitates is noted, positioned along grain boundaries, facilitating the observation of columnar grains developed during the AM. These grains exhibit a preferential direction, aligned with the vertical direction of wall fabrication, which coincides with the thermal diffusivity direction, appearing darker grey when examined under the microscope.

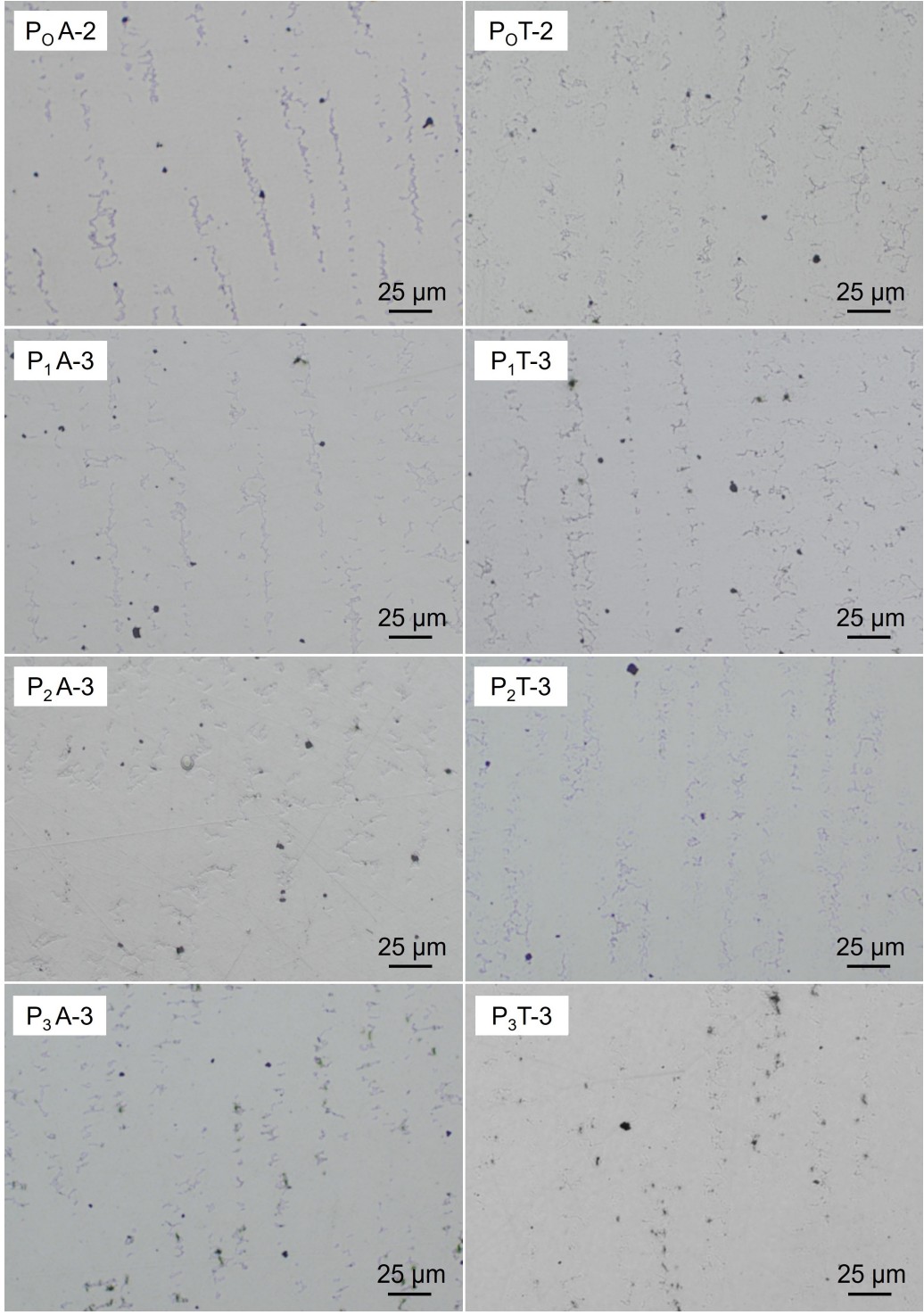

**Figure 5.** Optical microscopy (OM) images of specimens made from the four starting powders with and without thermal treatment.

For instance, the $P_O$ A-3 specimen was made from the original powder without thermal treatment and was located at the top (position 3) of the wall made from $P_O$. In the case of the other walls, the specimens were taken from center of each one, respectively. Regardless of the specimen position, the precipitates exhibited a clear orientation.

In general, in the untreated samples shown in Figure 5 (left, *A*), a well-defined matrix with precipitates concentrated at the grain boundaries was observed. When the thermal treatment was applied, as seen in Figure 5 (right, *T*), the precipitates did not fully dissolve

into the matrix. The thermal treatment decreased the size of the precipitates, making some of the precipitates rounder in shape. Following the treatment, the grains maintained their spherical morphology and columnar distribution. The distribution in the matrix was more homogeneous than without treatment.

Figure 6 shows CBS-SEM images taken at higher magnifications from specimens located in the center of each wall, showing the matrix and precipitates as well as their orientation under different manufacturing conditions and treatments.

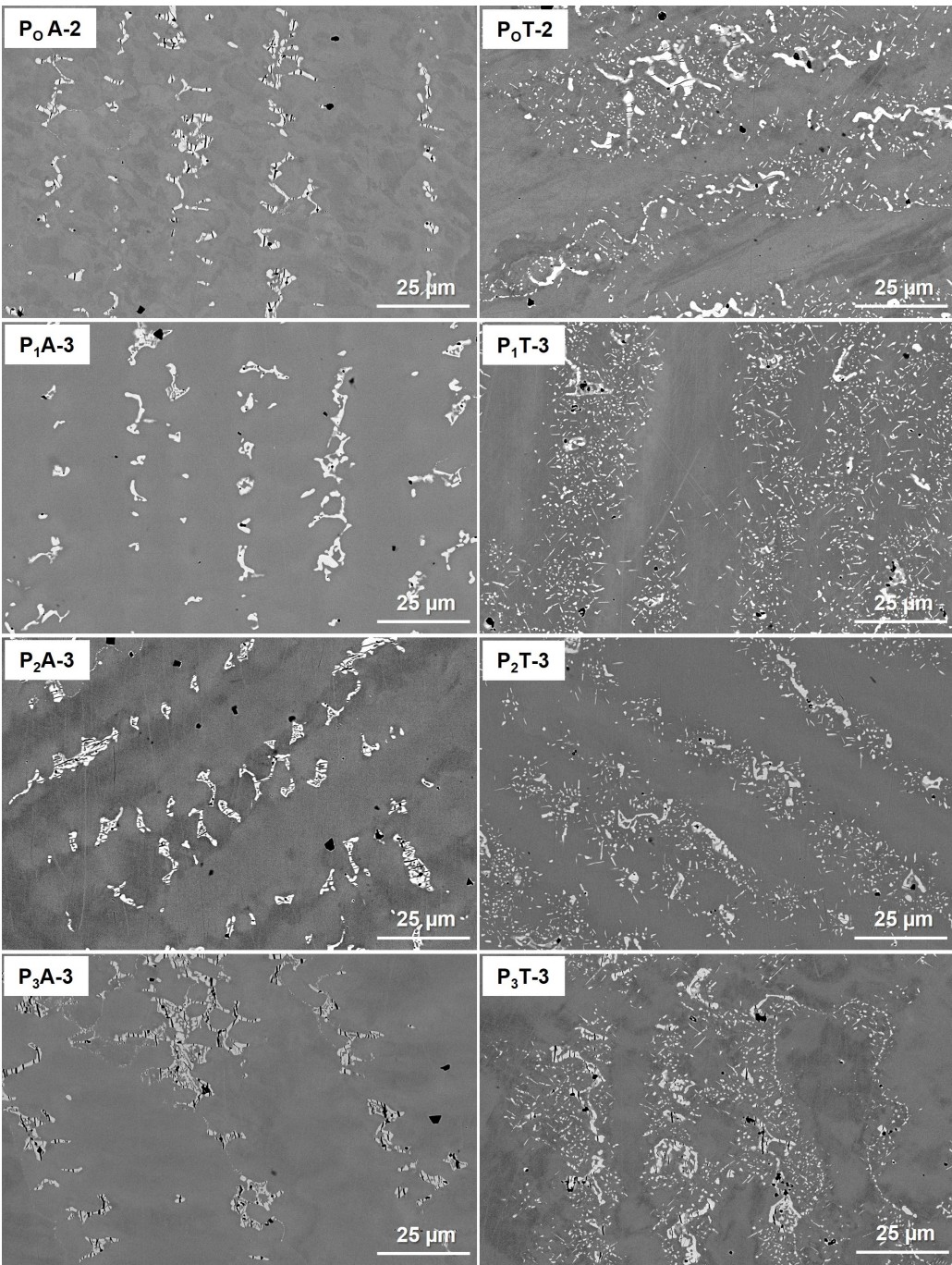

**Figure 6.** SEM images of specimens as-built (*A*) and with thermal treatment (*T*) made from the four types of starting powders.

The images in Figure 6 illustrate how the microstructures of the samples that were fabricated using a coarser starting powder size exhibited phases at the grain boundaries

with a greater extension. This observation was challenging to discern in the samples that had undergone the thermal treatment.

In the samples that underwent the thermal treatment, compared to those in their as-built state, a more extensive distribution and higher quantity of precipitates throughout the entire matrix (dark region) were appreciated, albeit of a smaller size. This finding aligns with the theory that thermal treatments facilitate the dispersion of precipitates and diffusion in the matrix of elements where the precipitate is rich. These precipitates remain, although with a reduced size, in the matrix and near the grain boundaries where residual precipitates persist. Furthermore, the size of the starting powder particles may also be a factor affecting the size of the aforementioned precipitates; specifically, the smaller the starting powder size ($d_{50}$), the smaller the precipitates sizes after the thermal treatment, as illustrated in Figure 7.

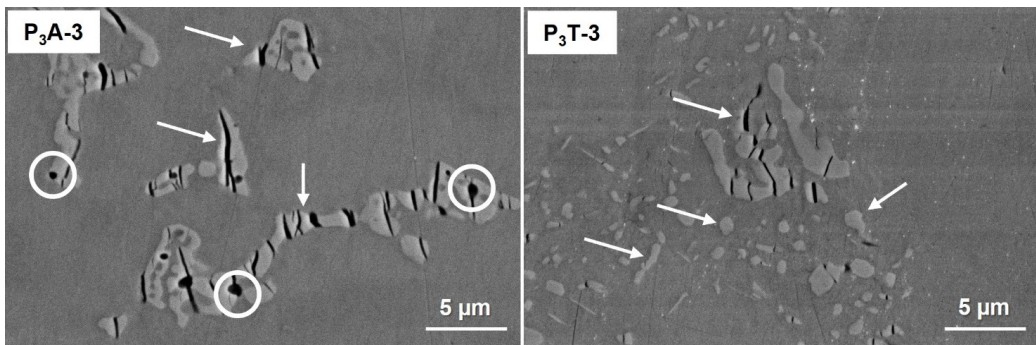

**Figure 7.** SEM images of the precipitate morphology before ($P_3$ A-3) and after thermal treatment ($P_3$ T-3).

The morphology of the precipitates was irregular and sinuous at the grain boundaries, being more clearly observed in the as-built samples.

The distribution and morphology of the precipitates exhibited slight variations after the thermal treatment. Furthermore, more porosity was observed in the specimens in the as-built condition (circles in Figure 7 left). This phenomenon occurred regardless of the starting powder used to build the four walls. It is important to mention that cracks (marked in Figure 7 with arrows) were detected in the precipitated phases due to shrinkage in both conditions. This can be clearly seen in Figure 7.

To determine the composition of the precipitates and their surrounding areas, EDS analyses were performed as mapping and spots at the most significant locations. Figure 8 confirms that the precipitates were rich in molybdenum, tungsten, and silicon.

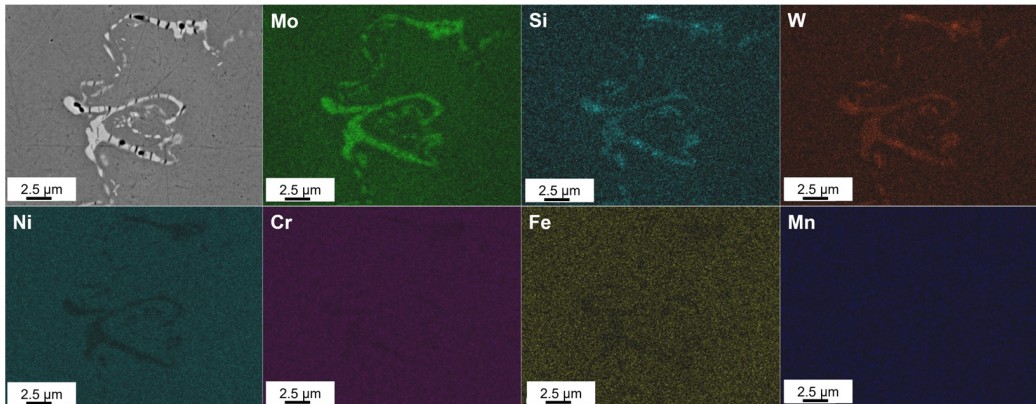

**Figure 8.** Mapping of the as-built specimens made from the original powders located in the center of the wall ($P_O$ A-2).

Figure 9 depicts the EDS analysis of key features within the metal matrix, including the analysis of several precipitates (spots 1 and 2) and the matrix composition (spot 3). Consistent with the mapping presented earlier, the analysis confirmed that the precipitates were of type M6C (where M represents molybdenum, chrome, or tungsten). According to the existing literature [28,29], these are commonly related to the alloying element contents exceeding the solubility limits. Furthermore, measurements at spot 3 revealed values that were similar to those of the standard material composition, with the exception of the molybdenum content, which was diffused into the precipitates. In the areas surrounding the precipitates, the diffusion mechanism was evident.

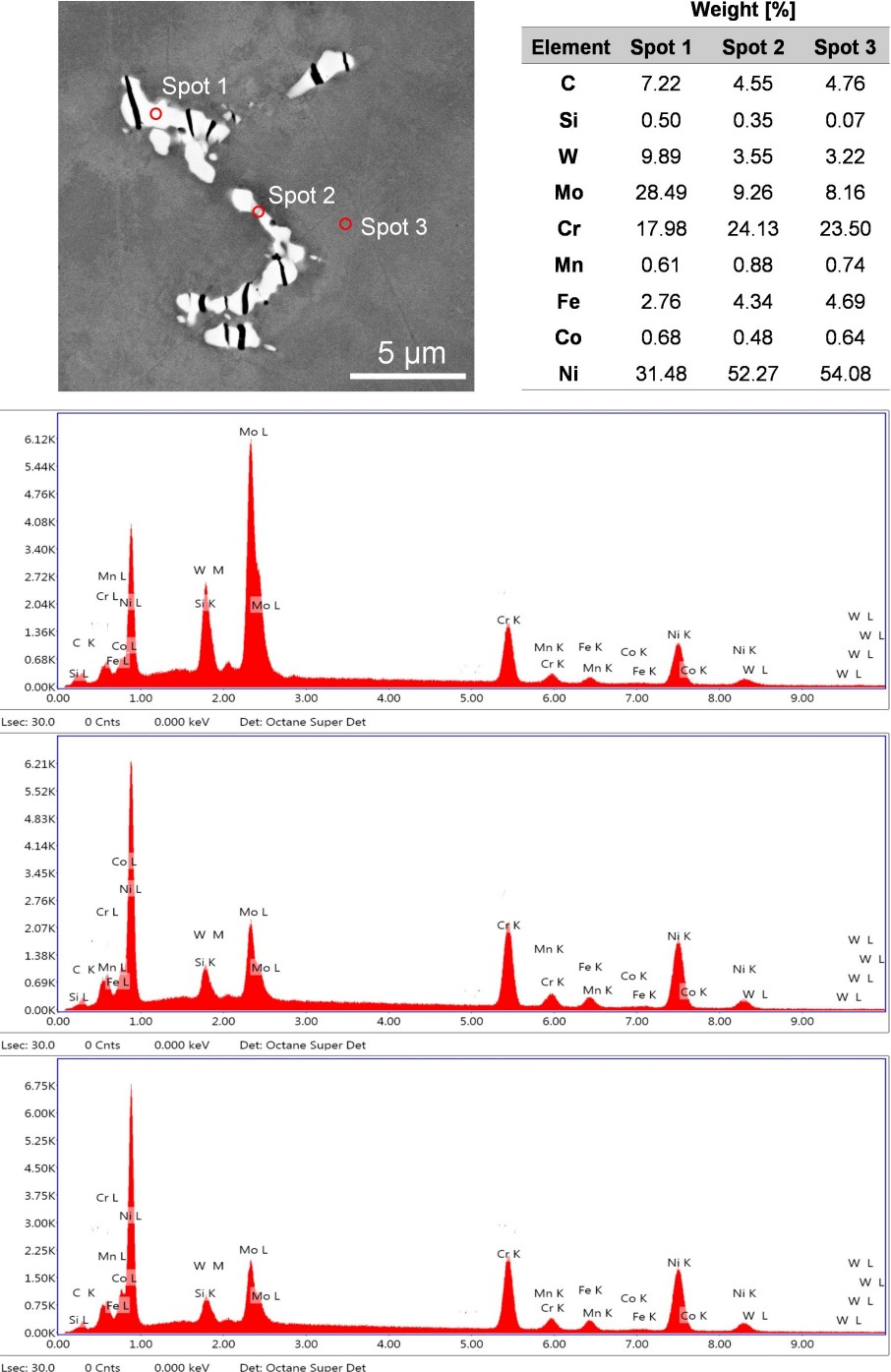

| | Weight [%] | | |
|---|---|---|---|
| Element | Spot 1 | Spot 2 | Spot 3 |
| C | 7.22 | 4.55 | 4.76 |
| Si | 0.50 | 0.35 | 0.07 |
| W | 9.89 | 3.55 | 3.22 |
| Mo | 28.49 | 9.26 | 8.16 |
| Cr | 17.98 | 24.13 | 23.50 |
| Mn | 0.61 | 0.88 | 0.74 |
| Fe | 2.76 | 4.34 | 4.69 |
| Co | 0.68 | 0.48 | 0.64 |
| Ni | 31.48 | 52.27 | 54.08 |

**Figure 9.** EDS analysis of most characteristic features in the metal matrix for the specimen $P_O$ A-2: precipitate analysis (spots 1 and 2) and matrix composition analysis (spot 3).

Figure 10 illustrates the X-ray analysis results of two samples. One sample was analyzed prior to undergoing the heat treatment, while the other specimen was analyzed post-heat treatment. A subtle disparity in the diffraction patterns was evident. Specifically, the sample subjected to heat treatment exhibited a discernible peak corresponding to a phase present in the precipitates that was enriched in molybdenum. Additionally, the peak of the alloy was notably sharper.

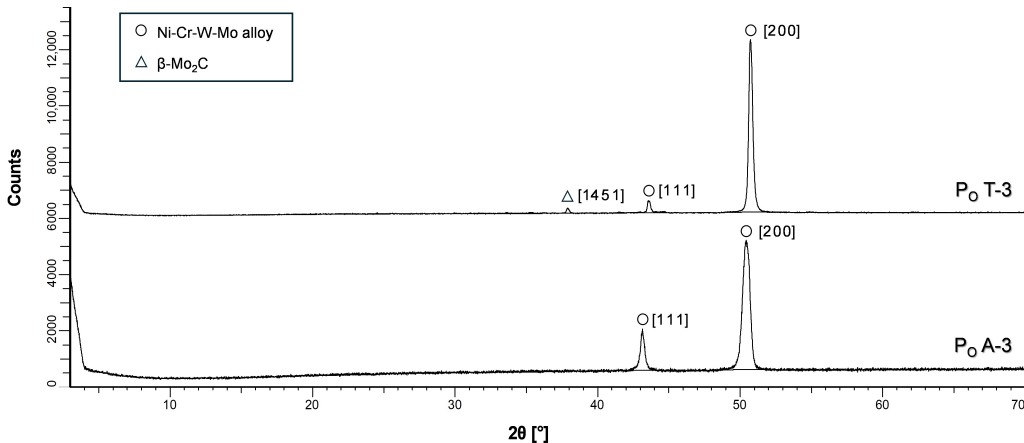

**Figure 10.** XRD analysis of specimens $P_O$ A-3 and $P_O$ T-3.

Conversely, the untreated specimen's diffraction pattern solely displayed peaks that were attributable to the alloy itself. These peaks, which were broader and more pronounced, result from molybdenum being part of the singular measured phase. This corroborates earlier findings derived from the SEM imagery and EDS analyses.

### 3.2. Mechanical Properties

The mechanical property results obtained from the tests are presented in Figure 11, showcasing the average values across the various specimens. The parameters analyzed included *UTS* in MPa, *YS* in MPa, and Young's modulus in GPa.

As anticipated, variations in the mechanical properties arose due to the thermal treatment. To further elucidate the influence of the initial powder composition and thermal treatment on these properties, several result comparisons are provided.

Within the framework of the starting powder employed to produce each wall (with granulometries designated as $P_O$, $P_1$, $P_2$, and $P_3$), some differences were observed in the final properties. The parameter that was most strongly affected by the variation in granulometry was the *UTS*, regardless of whether the specimens underwent the thermal treatment. The finer the powder employed, the smaller the *UTS* achieved; the specimens manufactured from $P_3$ (with a $d_{50}$ of 126.27 μm) showed higher *UTS* values, measuring 609 ± 32 MPa without the heat treatment and increasing to 744 ± 13 MPa with thermal processing, which represents the maximum average value. Hence, the samples made from the finest starting powder, $P_1$ (with a $d_{50}$ of 66.03 μm), exhibited the lowest *UTS* values, both before and after the thermal treatment (450 ± 40 MPa and 586 ± 88 MPa, respectively).

With regard to the *YS* values, there were no significant variations observed among the specimens made from the four starting powders, all hovering around 400 MPa in the specimens without heat treatment. Generally, this parameter increased slightly to around 440 MPa after the thermal treatment.

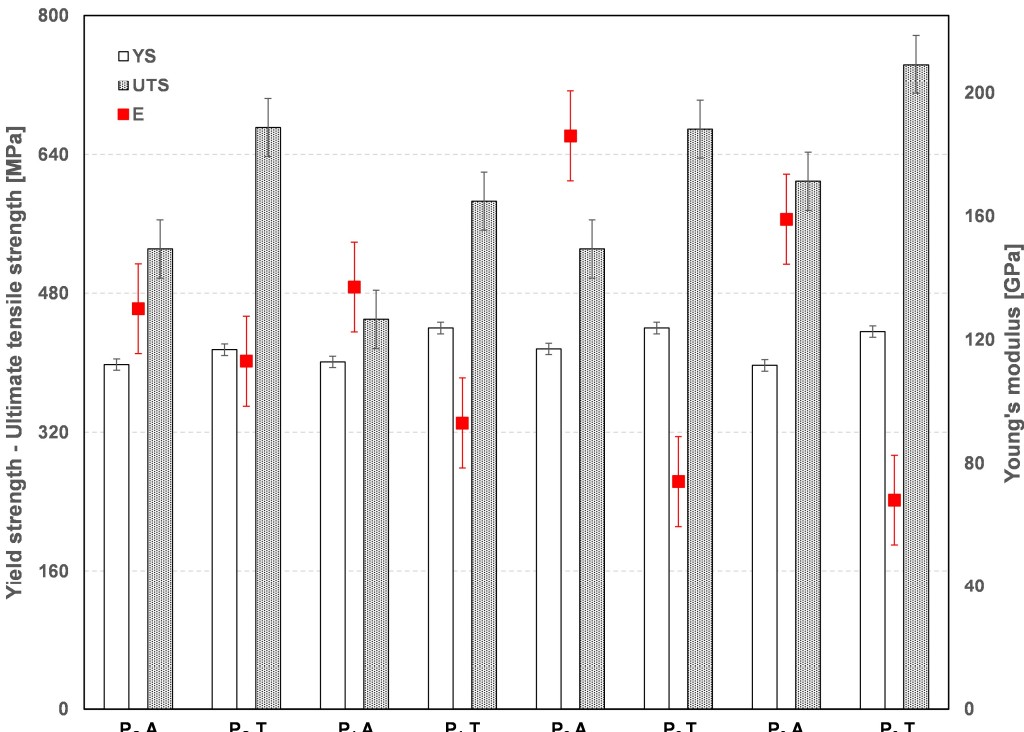

**Figure 11.** Results of the tensile tests conducted on the specimens of walls $P_O$, $P_1$, $P_2$, and $P_3$.

There was a general reduction in the Young's modulus of the specimens after they were thermally treated. The specimens made from $P_2$ (with a $d_{50}$ of 80.71 μm) without the thermal treatment stood out for having the highest values of Young's modulus (186 ± 21 GPa). For the specimens made from $P_3$ (with a $d_{50}$ of 126.27 μm), the diminution of this parameter was more than 50%, dropping from 159 ± 27 GPa in the as-built condition to 68 ± 4 GPa after the treatment.

The precipitates were situated at the boundaries of the columnar grains, making them more defined. As observed, the constituents forming the precipitates diffused into the matrix as a result of the thermal treatment. The heat treatment induced alterations in the precipitates, evidenced by the reduction in their size and the increase in their numbers. These changes affected the mechanical properties of the specimens. When the precipitates were more evenly distributed, the matrix hardened, increasing both the *YS* and the *UTS* values.

### 3.3. Instrumented Microindentation and Hardness

The results of the load (N) vs. penetration (μm) analyses are shown in Figure 12. The microindentation tests were performed on specimens placed in the middle height of each wall. The loads applied were 1, 5, and 10 N. The test results showing the relative elastic recovery are given in Figure 13. Independently of the applied load, the specimens made from the coarsest powder ($P_3$) showed less recuperation in comparison to the others. At the lowest load (1 N), the finer the starting powder used, the higher the penetration observed (Figure 12). This phenomenon was more noticeable after the thermal treatment.

In Figure 13, it is observed that there was a decrease in the relative elastic recovery as the size of the powder increased. This phenomenon was observed for both sample conditions: as-built and after thermal treatment. Furthermore, the lower the applied load, the higher the relative elastic recovery. The specimens that were thermally treated presented a higher relative elastic recovery than the ones that were as-built for the same starting powder size.

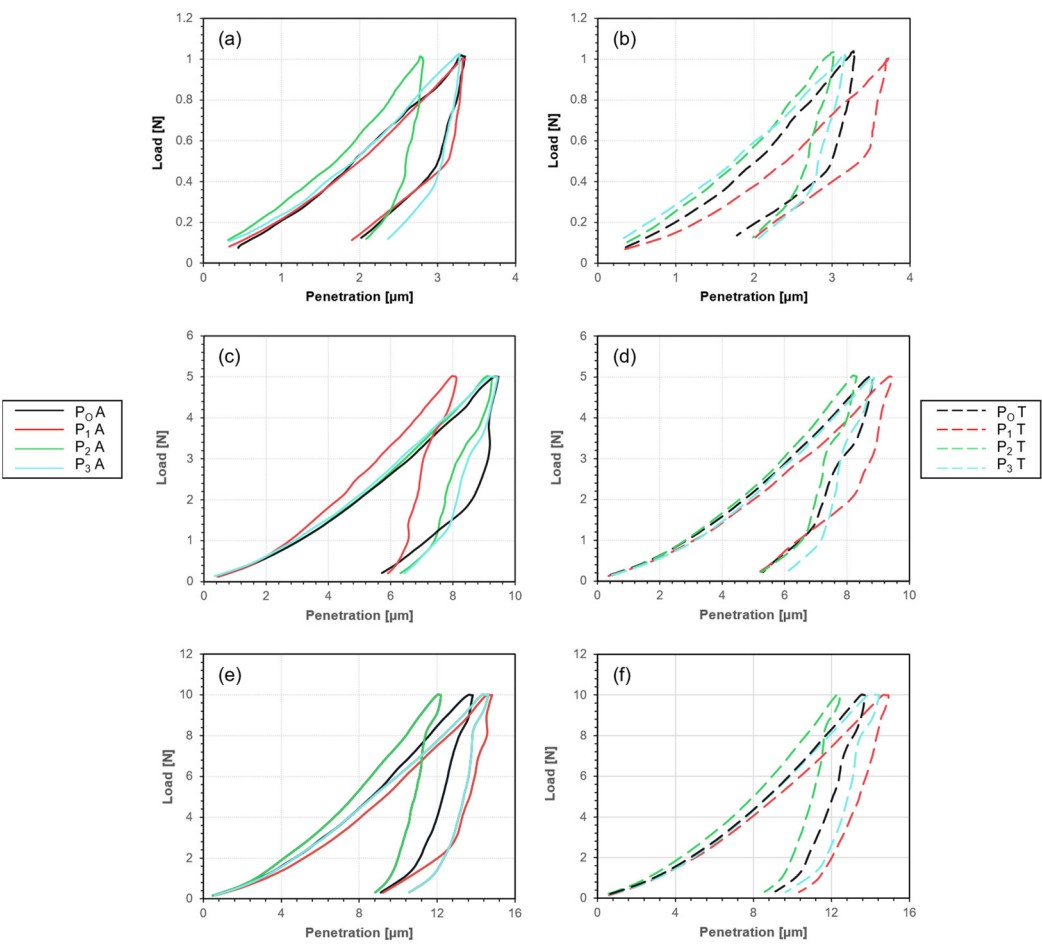

**Figure 12.** P–h curves for as-built specimens (**left**) and those with thermal treatment (**right**) at different loads: (**a**,**b**) 1 N; (**c**,**d**) 5 N; (**e**,**f**) 10 N.

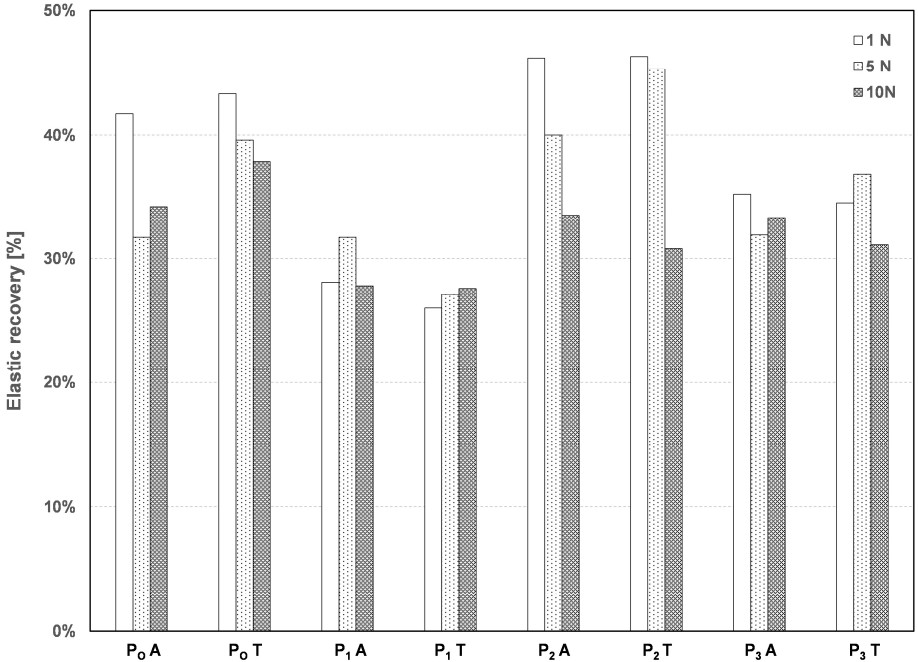

**Figure 13.** Results of the relative elastic recovery of the specimens.

An average of the Vickers hardness tests was obtained using three different load configurations for the microindentation test. The results are presented in Table 4.

**Table 4.** Results of the Vickers hardness calculated for the specimens.

| HV [MPa] | $P_O$ | $P_1$ | $P_2$ | $P_3$ |
|---|---|---|---|---|
| **A** | $308 \pm 10$ | $266 \pm 25$ | $269 \pm 16$ | $330 \pm 29$ |
| **T** | $303 \pm 6$ | $300 \pm 6$ | $280 \pm 21$ | $341 \pm 44$ |

The measured hardness values displayed a certain variability when assessing the specimens that were subjected to different starting powders and thermal treatments. This phenomenon arose from the intricate distribution of the precipitates within the matrix.

The wall with the coarsest powder size exhibited the highest Vickers hardness, which may be attributed to the size and morphology of the precipitates observed in the microstructure analysis.

Concerning the effects of the thermal treatment on the Vickers hardness values, there was a slight increase in the hardness of the treated specimens. This could be attributed to the greater dispersion of hard, rich-Molybdenum precipitates throughout the matrix, as illustrated in Figure 7.

### 3.4. Tribology—Wear

The wear coefficient serves as a metric that indicates a material's propensity for degradation when it is in contact with another material and exposed to frictional forces. A high wear coefficient indicates that the material is inclined to undergo rapid degradation under such conditions.

Microscopic photos were taken of the grooves made by the indenter during the test, which are shown in Figure 14.

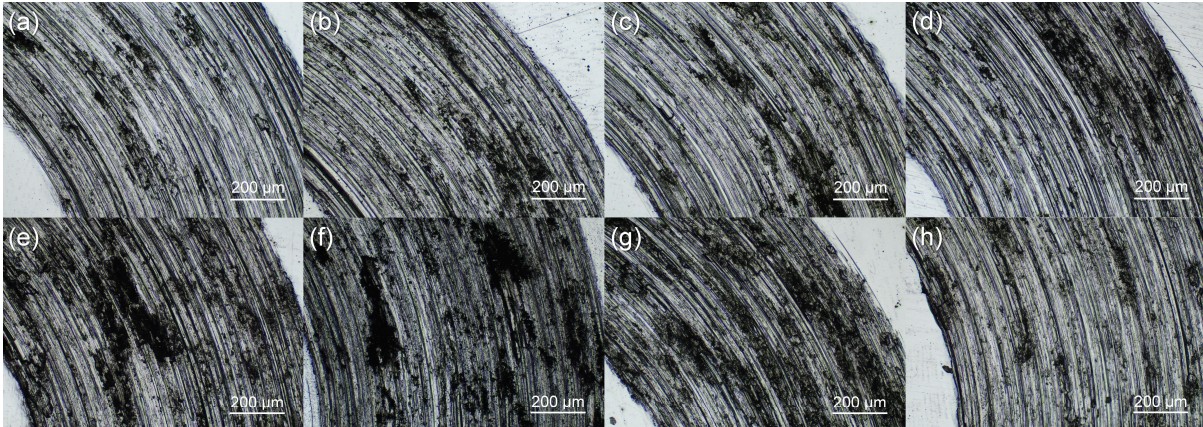

**Figure 14.** OM images showing the grooves created during the tribological ball-on-disk tests: (**a**) $P_O$ A-2, (**b**) $P_1$ A-3, (**c**) $P_2$ A-3, (**d**) $P_3$ A-3, (**e**) $P_O$ T-2, (**f**) $P_1$ T-3, (**g**) $P_2$ T-3, and (**h**) $P_3$ T-3.

The effects of the heat treatment were manifest in the tribological properties of the specimens. The grooves were wider in the specimens that were thermally treated, regardless of the starting powder size. This demonstrates a reduction in the wear properties as a result of the heat treatment.

Table 5 shows the mass losses experienced by each specimen after the tribology tests. According to these results, in the context of the starting powder granulometry, there was not a clear relationship between particle size and weight loss. On the other hand, more material was removed or detached when the specimens underwent the thermal treatment. This could be related to the softening of the matrix.

**Table 5.** Loss of mass (mg) suffered by each specimen after the tribology test.

| Mass Loss [mg] | $P_O$ | $P_1$ | $P_2$ | $P_3$ |
|---|---|---|---|---|
| **A** | 2.95 | 3.04 | 3.22 | 3.48 |
| **T** | 4.43 | 4.78 | 4.09 | 4.78 |

## 4. Conclusions

The main conclusions drawn are as follows:

- The samples fabricated using a coarser starting powder size displayed precipitates extending along the grain boundaries to a greater extent.
- Using a finer starting powder resulted in a decrease in the achieved *UTS*.
- There was a reduction in the relative elastic recovery as the size of the powder increased.
- The coarsest powder size yielded the highest Vickers hardness values.
- The thermal treatment led to the generation of finer-sized precipitates that were distributed more extensively throughout the entire matrix, consequently reducing the obtained Young's modulus values.
- The specimens subjected to the thermal treatment exhibited a higher relative elastic recovery, regardless of the starting powder size.
- Heat processing induced a slight increase in the Vickers hardness values, consistent with the enhanced dispersion of precipitates on the surface.
- Additionally, the heat treatment resulted in a reduction in the wear properties of the samples, evidenced by a lower mass loss but increased groove formation.

**Author Contributions:** Conceptualization, I.M.-M.; methodology, E.A.; formal analysis, C.A.; investigation, E.M.P.-S. and E.A.; resources, E.N. and M.K.; data curation, E.A.; writing—original draft preparation, E.M.P.-S.; writing—review and editing, I.M.-M.; supervision, C.A. All authors have read and agreed to the published version of the manuscript.

**Funding:** This research received no external funding.

**Data Availability Statement:** Data are contained within the article.

**Acknowledgments:** The authors want to thank the Universidad de Sevilla for the use of experimental facilitates at CITIUS, Microscopy and X-ray Laboratory Services (VII PPIT-2022-I.5 ISABEL MONTEALEGRE and PPIT-2023-I.5 CRISTINA ARÉVALO).

**Conflicts of Interest:** Enrique Ariza, Erich Neubauer, and Michael Kitzmantel are employed by the company RHP-Technology. The remaining authors declare that the research was conducted in the absence of any commercial or financial relationships that could be construed as a potential conflict of interest.

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
