# Peer review of "Manufacturing via Plasma Metal Deposition of Hastelloy C-22 Specimens Made from Particles with Different Granulometries"

_machines, doi:10.3390/machines12040253_

Round 1

Reviewer 1 Report

Comments and Suggestions for Authors

This manuscript reports the characterization of Hastelloy C-22 material grown by a plasma-assisted additive manufacturing method. The main goal consists in correlating microstructural, mechanical and tribological properties of the samples with two deposition variables, namely thermal treatment and granulometry. The topic is original, methodology is well explained and results are sound. Therefore, this contribution is suitable for publication in Machines.

Before acceptance, I suggest revision of many small items throughout the article. Please consider addressing the following points:

1. Abstract: please define UTS.

2. Table 2, granulometry: are so many decimal figures significant and necessary? Also, please define the granulometry parameters d_10, d_50 and d_90.

3. Fig. 1, SEM images of powder particles: Distribution in P0 resembles that of P3. There are very small particles in P3. Please comment.

4. Fig. 2, deposition equipment scheme: Add relevant sizes and distances. The regions occupied by plasma arc could be shown. What "Pilot gas" was used? The plasma atmosphere might modify the chemical composition of the powder by, for instance, functionalization of the particles. Please discuss.

5. Please add the flow rates set for the pilot gas and shielding gas.

6. What substrate was used?

7. How thick are the specimens cut for analysis? Besides cutting, did the specimens undergo surface polishing? This might impact microscopy analysis and wear/hardness tests.

8. Page 4, bottom: "The microstructural analysis was carried out after a metallographic preparation". Please describe the preparation and how surface finishing might reflect on OM and SEM observations.

9. Fig. 4, schemes of specimens: Provide height position of each specimen and total height of each wall. Fig. 4c: A-3 and T-3 are repeated.

10. Please specify the material and geometry of the indenter used in the microhardness measurements. Relative humidity in tribology tests should be provided.

11. Figs. 5 and 6, optical and SEM images: Indicate how the images were oriented with respect to the original sample. It will help to understand the distribution of the precipitates.

12. Fig. 9: Is it necessary to show the EDS spectra? If yes, please use bigger labels and mark the relevant peaks.

13. Fig. 10, please identify the sample of each XRD plot. Caption: check sample names.

14. Table 4, tensile tests: Check significant digits of the provided values.

15. Page 12, bottom: "...the finest the starting powder used, the higher penetration (figure 10)" --> figure 11.

16. Tribology - wear: Could you estimate the wear coefficient (wear rate?) and compare it with values from reference materials used in AM?

17. Fig. 12: Please add scale bars and indicate sample number for each tribology test. Also: "It can be observed that the finer precipitates generated after heat treatment are dislodged during wear tests" --> This claim is not clear from the optical images, please justify.

18. "The finer the powder, the less material loss; this is confirmed in Table 8": It does not seem consistent with granulometry data from Table 2 and Fig. 1 (SEM images).

19. What was done with the debris after each ball-on-disk tests? Were they removed or left in the groove? How was the debris distribution on the ball of the tribometer?

Author Response

REVIEWER #1 COMMENTS AND SUGGESTIONS

We thank the referee for his/her careful reading of our manuscript machines-2923101. Responses to specific comments are listed below:

R1C: Abstract: please define UTS.

A.R.1: Thank you for your suggestion. It has been included the definition in the text.

Page 1, Lines 21-22 “It was observed that the particle size slightly affects the final properties; the finer the powder, the lower the ultimate tensile strength values.”

R1C: Table 2, granulometry: are so many decimal figures significant and necessary? Also, please define the granulometry parameters d_10, d_50 and d_90.

A.R.2: Thank you for your observation. Indeed, there is no need in so many decimals; the definition of the parameters has been detailed in the manuscript.

Page 2, Lines 75-80 “Table 2 presents the nomenclature used to identify the different powders and walls, along with the sizes d10, d50, and d90 (d10 denotes the particle diameter at which 10% by weight of the particles are smaller than this diameter, while d50 and d90 provide analogous measurements for 50% and 90%). Those values were measured using a Mastersizer 2000 particle size analyser (Malvern Instruments Ltd, Malvern, Worcestershire, UK).”

R1C: Fig. 1, SEM images of powder particles: Distribution in P0 resembles that of P3. There are very small particles in P3. Please comment.

A.R.3: As you referred in your comment, these images are quite similar. Therefore, we have prepared a new collage of images for Figure 1.

Figure 1. SE-SEM images of the starting raw material Hastelloy C-22, in its original batch (PO), as well as in the form of fine powder (P1), medium powder (P2), and coarse powder (P3).

R1C: Fig. 2, deposition equipment scheme: Add relevant sizes and distances. The regions occupied by plasma arc could be shown. What "Pilot gas" was used? The plasma atmosphere might modify the chemical composition of the powder by, for instance, functionalization of the particles. Please discuss. Please add the flow rates set for the pilot gas and shielding gas.

A.R.4: Thank you for your comments. As this work is the continuation of another, referenced in the bibliography, multiple data settings have not been mentioned in this manuscript.

The distance between the torch and the substrate (or specimen) during the fabrication was 10 mm. The pilot gas was argon, with a flow rate of 1.5 L/min).

Working in an air atmosphere makes it necessary to employ another gas to act as a shield (shielding gas), with a flow rate of 15 L/min. In this investigation, pure argon (99.99% purity) was employed.

The material was fed as powder to the plasma jet by aligned holes. For improving the powder flow through the ducts, argon was injected (powder gas), with a flow rate of 1.5 L/min.

Regarding the possible modifications, it has been proved in previous authors’ works that there are no chemical modifications or functionalization of the particles due to the plasma atmosphere [26].

  1. Perez-Soriano, E.M.; Ariza, E.; Arevalo, C.; Montealegre-Melendez, I.; Kitzmantel, M.; Neubauer, E. Processing by Additive Manufacturing Based on Plasma Transferred Arc of Hastelloy in Air and Argon Atmosphere. Metals 2020, 10, 200. Doi: 10.3390/met10020200

The manuscript has been implemented as follows:

Pages 3-4, lines 94-103: “The manufacturing process involved several parameters, selected from previous authors’ research [26]. The energy source operated at a current intensity of 130 A, while the torch was programmed to move at a speed of 700 mm/min with oscillating movements. The deposition flow rate was 22.5 g/min. For improving the powder flow, argon was injected (powder gas), with a flow rate of 1.5 L/min. During the process, the distance between the torch and the built wall was 10 mm. Additionally, argon was introduced around the plasma arc to locally protect the weld seam from oxidation or other external agents that could enter during the manufacturing process; it is named shielding gas, with a 99.99% purity and a flow rate of 15 L/min. These parameters remained identical for the four walls; the resulting samples can be observed in Figure 3.”

R1C: What substrate was used?

A.R.5: The substrate material was AISI 1015, previously cleaned by brushing.

R1C: How thick are the specimens cut for analysis? Besides cutting, did the specimens undergo surface polishing? This might impact microscopy analysis and wear/hardness tests. Page 4, bottom: "The microstructural analysis was carried out after a metallographic preparation". Please describe the preparation and how surface finishing might reflect on OM and SEM observations.

A.R.6: Thank you for your question. This info has been already published along with previous other authors’ works.

  1. Perez-Soriano, E.M.; Ariza, E.; Arevalo, C.; Montealegre-Melendez, I.; Kitzmantel, M.; Neubauer, E. Processing by Additive Manufacturing Based on Plasma Transferred Arc of Hastelloy in Air and Argon Atmosphere. Metals 2020, 10, 200. Doi: 10.3390/met10020200

 The dimensions of the specimens are shown in the following figure:

Before the tests were performed, the specimens were grounded, polished, cleaned with acetone in an ultrasonic bath, and dried. All the mentioned tests (microscopy, wear, and hardness) were made on the paddle of the test sample, location chosen to avoid possible interferences with the data obtained from the tensile tests, which could provoke the early fracture of the sample as a result of weakening.

R1C: Fig. 4, schemes of specimens: Provide height position of each specimen and total height of each wall. Fig. 4c: A-3 and T-3 are repeated.

A.R.7: Thank you very much for noticing the mistake in the figure lettering. The image has been changed in the manuscript.

The minimal height of each wall was 34 mm, 42 mm, 44 mm, and 61 mm, respectively for PO, P1, P2, and P3. Specimens X-1 were obtained in height from 1 mm to 11 mm, X-2 from 11 mm to 21 mm, and so on.

R1C: Please specify the material and geometry of the indenter used in the microhardness measurements. Relative humidity in tribology tests should be provided.

A.R.8: The indenter is a diamond in the form of a square-based pyramid, internal angle of 136°. The relative humidity in the lab is typically 30%-35% with a room temperature between 20 and 25°C. It has been included in the text.

Page 5, Lines 145-149 “The tribological test was performed at room temperature (relative humidity in the range of 30%-35%) with a ball-on-disc Microtest MT/30/NI tribometer (Microtest, Madrid, Spain). The ball of the tribometer, with 6 mm-diameter, is alumina and applies with a force of 5 N for 15 minutes. The radius of the groove made was 2 mm and the rotational speed was set to 200 rpm.”

R1C: Figs. 5 and 6, optical and SEM images: Indicate how the images were oriented with respect to the original sample. It will help to understand the distribution of the precipitates.

A.R.9: The images are taken on the surface of the probes, specifically on the paddle as commented previously, but not with a specific orientation.

R1C: Fig. 9: Is it necessary to show the EDS spectra? If yes, please use bigger labels and mark the relevant peaks.

A.R.10: It is indeed certain that the labels were difficult to read. We have improved the figure and added to the manuscript.

Figure 9. EDS analysis of most characteristics features in the metal matrix for the specimen PO A-2: Precipitates analysis (Spot 1 and 2) and matrix composition analysis (Spot 3).

R1C: Fig. 10, please identify the sample of each XRD plot. Caption: check sample names.

A.R.11: Thank you again for catching the mistake. It has been solved, both in the figure and its caption.

Figure 10. XRD analysis of specimens PO A-3 and PO T-3.

R1C: Table 4, tensile tests: Check significant digits of the provided values.

A.R.12: Figure 11 summarize the results of the tensile tests, following the suggestions of another reviewer.

Figure 11. Results of the tensile tests conducted on specimens of walls PO, P1, P2, and P3.

R1C: Page 12, bottom: "...the finest the starting powder used, the higher penetration (figure 10)" --> figure 11.

A.R.13: It has been changed, thanks a lot.

R1C: Tribology - wear: Could you estimate the wear coefficient (wear rate?) and compare it with values from reference materials used in AM?

A.R.14: These values were deleted, due some problems with the device. Some starting points of the plots were unstable; therefore, they couldn’t be trusted. If you consider that we have to include these graphics, it will be done.

R1C: Fig. 12: Please add scale bars and indicate sample number for each tribology test. Also: "It can be observed that the finer precipitates generated after heat treatment are dislodged during wear tests" --> This claim is not clear from the optical images, please justify.

A.R.15: Thank you for your suggestion. Scales have been included in the corresponding figure.

Figure 14 (old 12). OM images showing the grooves created during the tribological ball-on-disk tests: (a) PO A-2, (b) P1 A 3, (c) P2 A-3, (d) P3 A-3, (e) PO T-2, (f) P1 T-3, (g) P2 T-3, and (h) P3 T-3.

In the text, we have deleted the sentence you refer, which could lead to confusion and misunderstanding. We could mean that this debris could possibly be related with heat-treated samples.

R1C: "The finer the powder, the less material loss; this is confirmed in Table 8": It does not seem consistent with granulometry data from Table 2 and Fig. 1 (SEM images).

A.R.16: Thank you for your observation. We agree that this paragraph can cause a misunderstanding. It could be a slight relationship between the particle size and the mass loss, but it is indeed not clear. In order to avoid this misunderstanding, we have deleted this paragraph and introduce another one.

Page 13, Lines 320-324. “Table 5 shows the mass losses experienced by each specimen after the tribology tests. According to these results, in the context of the starting powder granulometry, there is not a clear relationship between particle size and weight loss. On the other hand, more material is removed or detached when the specimens undergo thermal treatment; this could be related with the softening of the matrix.”

R1C: What was done with the debris after each ball-on-disk tests? Were they removed or left in the groove? How was the debris distribution on the ball of the tribometer?

A.R.17: The samples were cleaned after the test. The ball is clean after running the experiments, the debris stays in the specimens, is not adhered to the ball.

We hope it is found satisfactory. Best regards,

Reviewer 2 Report

Comments and Suggestions for Authors

The manuscript investigates the influence of starting powder size and thermal treatment on the final properties of additively manufactured Hastelloy C-22 using Plasma Metal Deposition. Overall, the manuscript is well-structured, utilizes interesting research areas, and presents results and conclusions comprehensively. Following revisions are suggested to enhance clarity and presentation:

·         Emphasize the novelty of the work more explicitly.

·         Consider illustrating comparisons from "Section 3.2 (Mechanical properties)" using diagrams rather than tables to enhance reader comprehension.

·         Table 6 could be supplemented with a diagram too, for improved visual representation.

Author Response

REVIEWER #2 COMMENTS AND SUGGESTIONS

We thank the referee for his/her careful reading of our manuscript machines-2923101. Responses to specific comments are listed below:

R2C: Emphasize the novelty of the work more explicitly.

A.R.1: Thank you for your comment. The novelty of this work to analyse and determinate if there is a correlation between the particle size and the final behaviour of specimens produced via additive manufacturing. In the field of 3D printing, there aren’t studies where this relationship is investigated in a clear way. The manufacturing process, in this case Plasma Metal Deposition, is also another approach to develop specimens.

One sentence has been included in the abstract to showcase the value of our research, as you suggested.

Page 1, Lines 14-16 “The novelty of this work to analyse and determinate if there is a correlation between the particle size and the final behaviour of specimens produced via additive manufacturing.”

R2C: Consider illustrating comparisons from "Section 3.2 (Mechanical properties)" using diagrams rather than tables to enhance reader comprehension.

A.R.1: We have considered your comment, therefore, we have replaced Table 4 with Figure 11.

Figure 11. Results of the tensile tests conducted on specimens of walls PO, P1, P2, and P3.

R2C: Table 6 could be supplemented with a diagram too, for improved visual representation.

A.R.1: As well as we proceed with your previous comment, we have replaced Table 6 with Figure 13.

Figure 13. Results of the relative elastic recovery of the specimens.

We hope it is found satisfactory. Best regards,

Round 2

Reviewer 2 Report

Comments and Suggestions for Authors

The manuscript is in a good form now. I recommend it for publication.